

# Age-related change in mammographic breast density of women without history of breast cancer over a 10-year retrospective study

Aiko Ohmaru[1,2], Kazuhiro Maeda[3,4], Hiroyuki Ono[3,4], Seiichiro Kamimura[3,4,5], Kyoko Iwasaki[3,4], Kazuhiro Mori[3,4] and Michiaki Kai[6]

[1] Department of Environmental Health Science, Oita University of Nursing and Health Sciences, Oita, Japan
[2] Department of Radiological Science, Junshin Gakuen University, Fukuoka, Japan
[3] Station Clinic, Medical Corporation Shin-ai, Fukuoka, Japan
[4] Tenjin Clinic, Medical Corporation Shin-ai, Fukuoka, Japan
[5] Division of Total Health Care Unit, Chiyukai Shinkomonji Hospital, Fukuoka, Japan
[6] Nippon Bunri University, Oita, Japan

Corresponding author
Aiko Ohmaru, oumaru.a@junshin-u.ac.jp

## ABSTRACT

**Background.** Women with higher breast density are at higher risk of developing breast cancer. Breast density is known to affect sensitivity to mammography and to decrease with age. However, the age change and associated factors involved are still unknown. This study aimed to investigate changes in breast density and the associated factors over a 10-year period.

**Materials and Methods.** The study included 221 women who had undergone eight or more mammograms for 10 years (2011–2020), were between 25 and 65 years of age, and had no abnormalities as of 2011. Breast density on mammographic images was classified into four categories: fatty, scattered, heterogeneously dense, and extremely dense. Breast density was determined using an image classification program with a Microsoft Lobe's machine-learning model. The temporal changes in breast density over a 10-year period were classified into three categories: no change, decrease, and increase. An ordinal logistic analysis was performed with the three groups of temporal changes in breast density categories as the objective variable and the four items of breast density at the start, BMI, age, and changes in BMI as explanatory variables.

**Results.** As of 2011, the mean age of the 221 patients was $47 \pm 7.3$ years, and breast density category 3 scattered was the most common (67.0%). The 10-year change in breast density was 64.7% unchanged, 25.3% decreased, and 10% increased. BMI was increased by 64.7% of women. Breast density decreased in 76.6% of the category at the start: extremely dense breast density at the start was correlated with body mass index (BMI). The results of the ordinal logistic analysis indicated that contributing factors to breast density classification were higher breast density at the start (odds ratio = 0.044; 95% CI [0.025–0.076]), higher BMI at the start (odds ratio = 0.76; 95% CI [0.70–0.83]), increased BMI (odds ratio = 0.57; 95% CI [0.36–0.92]), and age in the 40s at the start (odds ratio = 0.49; 95% CI [0.24–0.99]). No statistically significant differences were found for medical history.

**Conclusion.** Breast density decreased in approximately 25% of women over a 10-year period. Women with decreased breast density tended to have higher breast density or

higher BMI at the start. This effect was more pronounced among women in their 40s at the start. Women with these conditions may experience changes in breast density over time. The present study would be useful to consider effective screening mammography based on breast density.

# INTRODUCTION

In Japan, the incidence rate of breast cancer has increased year by year (*Katanoda et al., 2021*), whereas the mortality rate has tended to level off since 2010. This contrasts with Western countries, where the mortality rate has leveled off or decreased since 2000 (*Henley et al., 2020*; *Huang et al., 2021*). Effective prevention of breast cancer is to identify women at increased risk. Although age, family history, reproductive factors, estrogen and lifestyle are established risk factors for breast cancer (*Britt, Cuzick & Phillips, 2020*), breast density has especially attracted attention. Globally, research on dense breasts began in the 1970s (*Wolfe, 1976*; *Wolfe, 1977*), and numerous studies have been reported since the 1990s (*Boyd et al., 1998*; *Warner et al., 1992*; *Szklo, Salane & Wolfe, 1991*). In Japan, there has been a high level of interest in dense breasts since around 2016 with the enactment of the Breast Density Notification Law in the US (*Kasahara, 2019*).

Breast densities fall into four categories by the ratio of mammary glands to fat in the breast tissue. The definition of a dense breast in Japan is a breast that is classified as Category 3 (heterogeneously dense) or Category 4 (extremely dense) according to the four breast density categories defined by the Breast Imaging Reporting and Data System (BI-RADS®) (*Mendelson, 2003*; *Japan Radiological Society & Japanese Society of Radiological Technology, 2021*). Dense breast is known to increase the risk of breast cancer and decrease the detection rate of lesions on mammographic images (*Boyd et al., 2007*; *Kerlikowske et al., 2007*; *Wong et al., 2011*). The case-control studies on dense breasts have been conducted in Japan, and a trend toward an increased risk of developing the disease has been reported (*Nagata et al., 2005*; *Kotsuma et al., 2008*; *Nishiyama et al., 2020*). Nishiyama et al. showed that the age-adjusted odds ratio (OR) for breast cancer was higher for category 4 (OR =2.12; 95% CI [1.28–3.49]) than for category 1 among Japanese women. *Lam et al. (2000)* and *Atakpa et al. (2021)* also revealed that breast density was a risk factor for breast cancer in postmenopausal women and those with a high body mass index (BMI).

The cross-sectional study found the percentages of breast density categories depended on the woman's age group. It has been thought that the mammary glands may shrink with age. *Warren et al. (2019)* reported that the average volume breast density (VBD) decreased by approximately 11% over six years but remained unchanged in 80% of women. In addition, another study showed a decrease in mammary density of approximately 11% over 10 years (*Lokate et al., 2013*). Both studies examined breast density over time for each woman, suggesting that breast density changes as women age. However, there is no

sufficient evidence on change of breast density with age. Regarding the association between breast cancer risk factors and breast density, age, BMI, and physical activity have been reported as determinants of breast density changes (*Azam et al., 2019*). In a longitudinal study of the effect of menopause on mammographic density, menopause reduced the area of dense tissue (*Boyd et al., 2002*).

Women with dense breasts should continue to undergo breast cancer screening because of their high risk of developing breast cancer. However, mammography of dense breasts is known to have a low lesion detection rate (*Carney et al., 2003*; *McCormack & Silva, 2006*). Therefore, a combination of mammography has been considered with other imaging techniques, such as breast ultrasonography or MRI (*Mariscotti et al., 2014*; *Ohuchi et al., 2016*). It would be necessary to consider the time and financial burden on women if multiple examinations are to be undergone.

Even in women with high breast density, the breast density can decrease over time. Suppose the rate and duration of decline in breast density, or the factors that predispose to a decline in breast density, can be determined. Estimating the age at which the dense breast changes to a fatty breast due to a decline in mammary gland density will be possible. Such estimates would help select the type of screening according to the subjects' risk factors.

The present study aimed to clarify the proportion of four breast density categories, investigate the changes in breast density over time, and understand which factors may cause those changes among Japanese women in a single health-screening facility.

## MATERIALS & METHODS

### Data description and study population

This is a retrospective cohort study in Japan. A total of 22,034 women who had mammography between April 2011 and March 2020 in Tenjin Clinic and Station Clinic, Medical Corporation Shin-ai were included. Only women who had undergone 8 or more mammograms and measured BMI over a 10-year period were eligible for the study. Also, the women were limited to ''no abnormality'' (including apparently benign disease) on all examinations to avoid mammography findings' influence on breast density. We categorized women to the presence or absence of a history of benign breast or gynecological diseases. None of the women had a history of breast cancer. In addition, the women were limited to between the ages of 25 and 65 years in 2011. 857 women had undergone 8 or more mammograms over a 10-year period. 669 women were ''no abnormality'' (including apparently benign disease) on all examinations. Of these, 221 women with eight or more BMI results, medical histories, and digital mammography data recorded since 2011 were included in the final study population (Fig. 1).

Data on mammography images, year of examination, age, BMI, and medical history were collected on the subjects. JPEG images of mammography examinations were obtained from the image server among the subjects' 10 years of examination data. Women's age, BMI, and medical histories were obtained from electronic medical records.

This study was approved by the Medical Corporation Shin-ai Ethics Committee (Acceptance Number: 5). Women obtained written consent to anonymize the data and

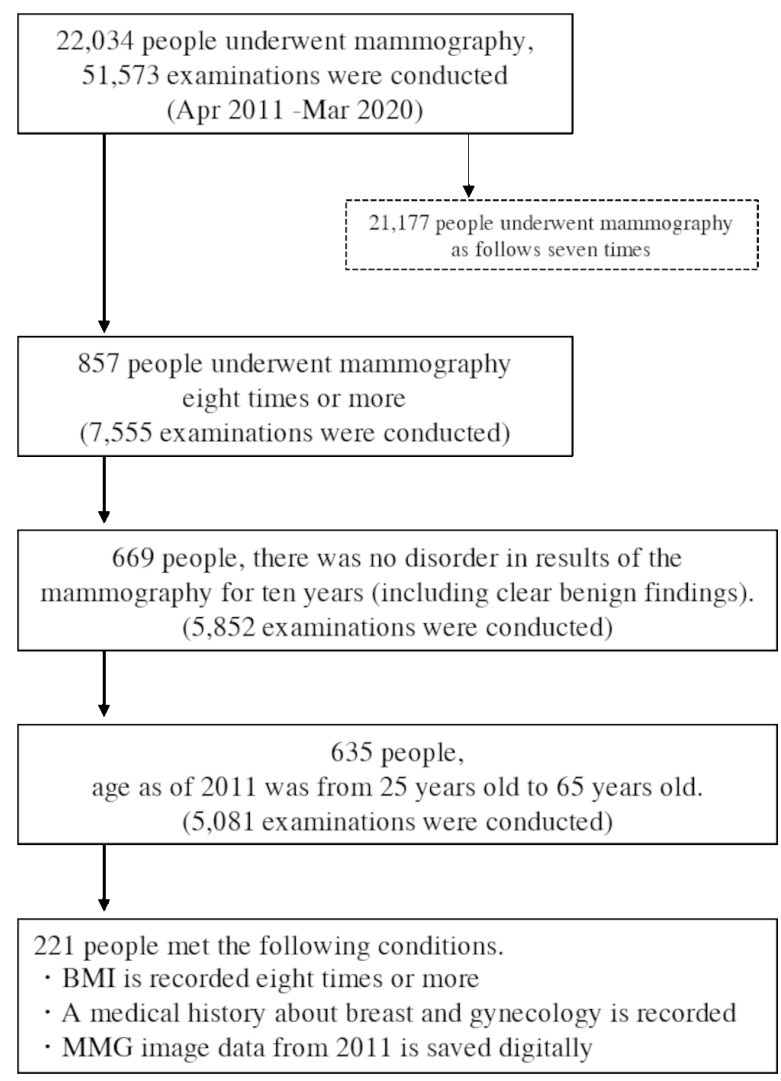

**Figure 1** Flowchart of selecting the retrospective women cohort who had a mammography in 2011–2020.

use it for research. Mammographic images were provided in JPEG format with personal information removed. Mammograms were imaged on Fujifilm AMULET and Canon (Toshiba) Peruru.

## Breast density assessment

The ratio of mammary glands and fat in the breast tissue can divide breast density into four types; fatty, scattered, heterogeneously dense, and extremely dense. This study used a machine-learning-based decision model to classify the following categories.
Category 1: Fatty
Category 2: Scattered
Category 3: Heterogeneously dense

Category 4: Extremely dense

The image data provided for this study were JPEG images for personal information protection. Commercially available BD quantification products could not be used for the study. Therefore, in order to consistently and objectively estimate the breast density of mammography from 2011 to 2020, a machine-learning model of image classification was developed. Since there is relatively little training data developing machine-learning models in the medical field, the training data can be augmented by data augmentation and transfer learning (*Ragab et al., 2019*). We used Lobe (*Microsoft, 2020*), a machine-learning model development software developed by Microsoft, which allows users to create their own judgment model by training arbitrary images on a model that has already trained a large number of images. A total of 695 normal images from the Digital Database for Screening Mammography (DDSM) database (*Heath et al., 1998*; *Heath et al., 2001*), which contains four breast composition categories with ACR image evaluation, were used to train the model.

Images loaded into Lobe are randomly divided into training and test images. These training images were used to create a machine-learning decision model. The accuracy of the decision model was able to be verified by the test images. Since the DDSM images were captured by scanning analog film, there would be images including those in which markers indicating the direction of capture were superimposed on the breast, or images that were partially filled in to hide personal information. Using the Lobe operation screen, we could identify images that may be confusing to create the decision model from among the training images that have been registered by dividing them into breast configuration categories. These identified images were manually improved by trimming an improper part or removed if it would bring an inappropriate learning. The modified image was replaced with the initial image. Even after the images were improved, the images were removed if Lobe might indicate a wrong category. By repeating these operations step by step, the accuracy of the model trained with DDSM images improved to 97%. This machine-learning model determined the category with the highest probability of being the most applicable of the four breast density categories. The trained model was exported and incorporated into our developed program. Using this program, JPEG mammographic images retrieved from the image server were classified into one of four breast-density categories (Fig. 2).

Mammographic images were determined individually for each of the right and left breasts. The results of breast density categories were output in csv files. The results of determination of breast density categories were recorded separately for the left and right breasts. Based on whether breast density categories changed, the changes in breast density were classified into three categories: Decrease, No change, and Increase. Similarly, based on whether BMI changed over time, the changes in BMI were classified into three categories: Decrease, No change, and Increase.

## Statistical analysis

Breast density category, BMI, age, and history were tabulated by women on each examination. Fisher's probability test was performed on the relationship between age at the beginning and breast density. Similarly, Fisher's probability test was performed for
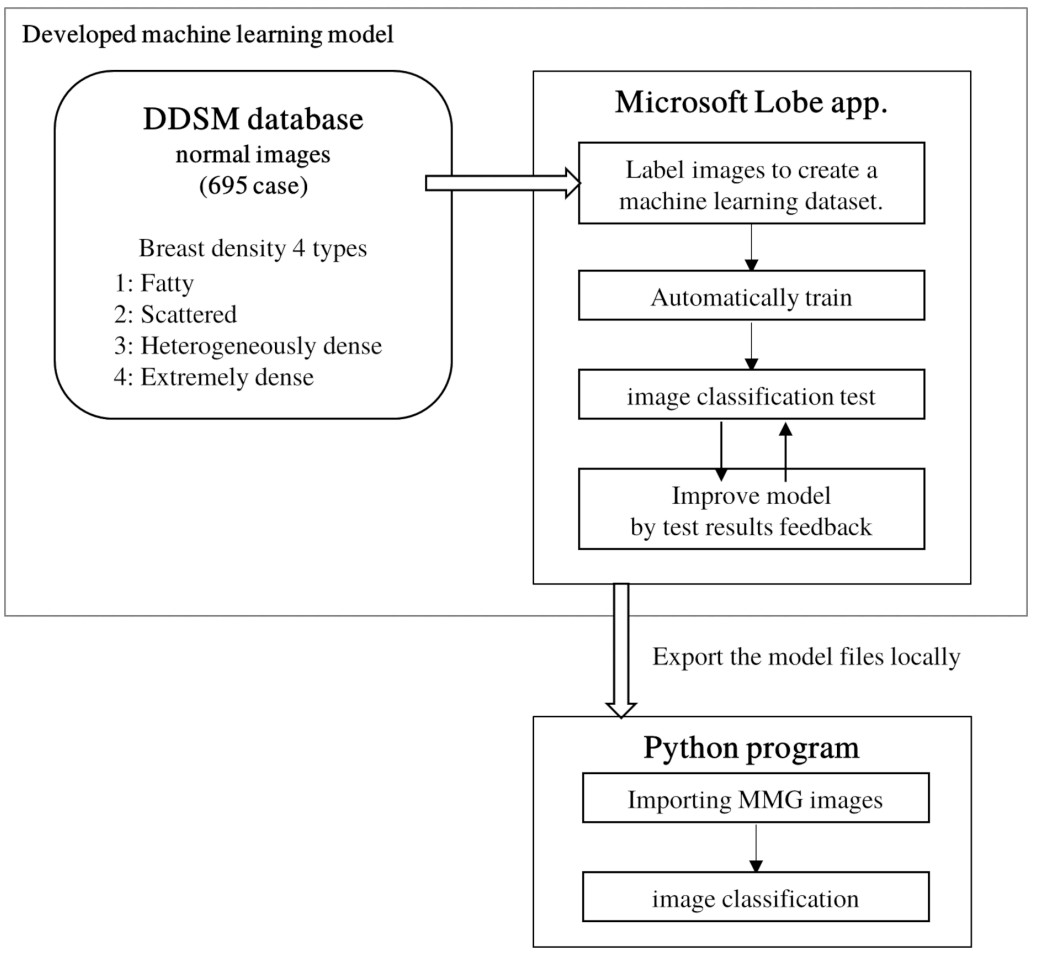

**Figure 2 Development of a classification model using machine learning to classify mammography images into breast density configurations.** Lobe (Microsoft, Redmond, WA, USA) was used to develop the model. Model tests and feedback were in Lobe's function.

the changes in breast density categories. Finally, an ordinal logistic analysis was performed with the three groups of temporal changes in breast density categories as the objective variable and the five items of breast density at the start, BMI, age, changes in BMI, and the presence or absence of a history of benign breast or gynecological diseases as explanatory variables. Only BMI at the start was used as a continuous variable. Since this study's objective variable was ordinal, we conducted an ordinal logistic analysis to analyze the factors associated with changes in breast density. Statistical analysis was performed using the statistical analysis free software R (*R Core Team, 2022*).

## RESULTS

Table 1 summarizes the results of BMI, history of the disease, and breast density determination at the starting age. There were 221 subjects with age of 47 $\pm$ 7.3 years and BMI of 21.3 $\pm$ 3.0 in 2011. The largest group of subjects were women in their 40s

**Table 1** Baseline characteristics in 2011 for the 221 women included in the study, separated by BMI, medical history, and breast density category at the starting age group.

| | | Starting age (years), no. (%) | | | |
|---|---|---|---|---|---|
| | Total No. (%) | <40 | 40–49 | 50–59 | ≧60 |
| No. of woman | 221 | 27 | 120 | 60 | 14 |
| BMI (kgm-2) | 21.3 ± 2.99 | 21.5 ± 4.12 | 21.2 ± 2.57 | 21.7 ± 3.15 | 21 ± 3.02 |
| Medical History of benign breast or gynecological diseases | | | | | |
| Absence | 146 (66.1%) | 25 (92.6%) | 77 (64.2%) | 38 (63.3%) | 6 (42.9%) |
| Presence | 75 (33.9%) | 2 (7.4%) | 43 (35.8%) | 22 (36.7%) | 8 (57.1%) |
| Breast density category | | | | | |
| 1 | 2 (0.5%) | 1 (1.9%) | 0 (0%) | 1 (0.8%) | 0 (0%) |
| 2 | 97 (21.9%) | 8 (14.8%) | 51 (21.3%) | 31 (25.8%) | 7 (25%) |
| 3 | 296 (67.0%) | 29 (53.7%) | 167 (69.6%) | 81 (67.5%) | 19 (67.9%) |
| 4 | 47 (10.6%) | 16 (29.6%) | 22 (9.2%) | 7 (5.8%) | 2 (7.1%) |

**Notes.**

BMI, body mass index.

(120 women). The results showed that category 3: heterogeneously dense, was the most common (67.0%, 269 cases), followed by category 2: scattered (21.9%, 97 cases), category 4: extremely dense (10.6%, 47 cases), and category 1: fatty (0.5%, 2 cases).

Figure 3 shows the proportion of breast density categories by age at the start. The horizontal axis is the starting age, and the vertical axis is the percentage of each breast density. The proportion of category 4, classified as dense breast, is the highest among those under the 40s ($p = 0.001$). Multiple comparisons by Fisher's test for each age group showed significant differences between those under 40 and 40s ($p = 0.001$) and 50s ($p = 0.001$). No significant difference was found because of the small number of subjects in the 60s.

Table 2 shows changes in breast density categories and BMI over the 10 years from 2011 to 2020. To focus on the final breast density, the breast density at the start of screening and in the final year was compared without considering slight variation changes during the period. The breast density categories were tabulated separately for the left and right breasts. For the entire subject population, 64.7% (286 cases) had no change in the breast density category, followed by a decrease of 25.3% (112 points) and an increase of 10% (44 patients). BMI increased in 143 (64.7%), decreased in 69 (31.2%), and showed no change in 18 (4.1%). Figure 4 shows the category changes compared to breast density categories at the start of the screening. The largest proportion of breast density change was in category 4 at 76.6%. The highest group with no change in the breast density category was in category 3 at 74.3%, and the lowest proportion was 23.4% in category 4. Fisher's probability test on the data in Fig. 4 showed a statistically significant difference ($p < 0.005$), indicating a difference in the distribution of each category ($p = 2.2E-16$). There was also a statistically significant difference between the two groups in categories 1 and 4 ($p = 0.0102$). Figure 5 shows the relationship between breast density category at the start and BMI at the start. BMI at the start correlated with breast density category, with BMI in category 1 (95% confidence intervals (CI): 26.7–28.9) being significantly higher than in category 2 (95% CI [23.4–23.8]), category 3 (95% CI [21.1–21.3]), and category 4 (95% CI [18.3–18.8]).

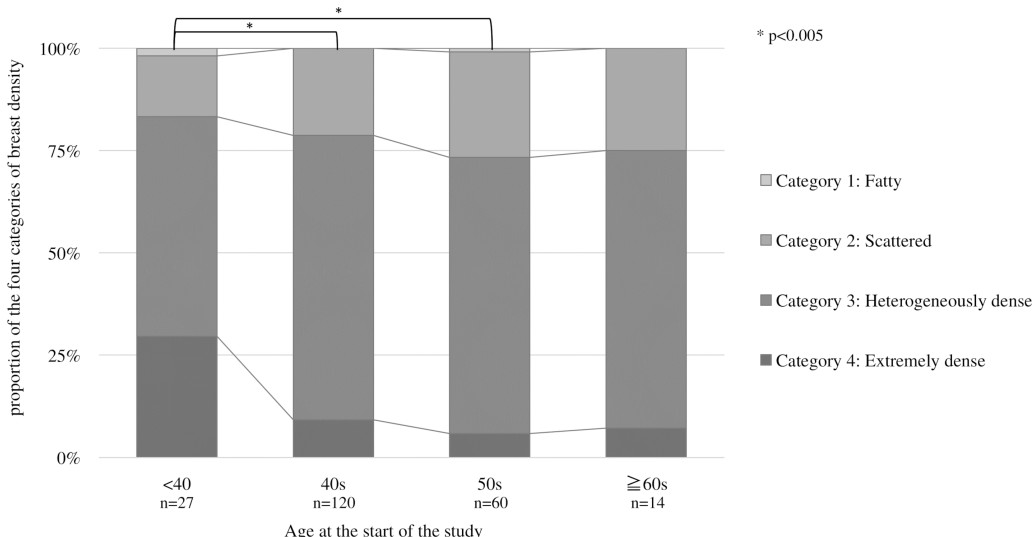

**Figure 3** **Proportion of the four categories of breast density by starting age at the start.** For each age group, multiple comparisons by Fisher's test showed significant differences between those under 40 and 40s ($p = 0.001$) and 50s ($p = 0.001$).

**Table 2** **Comparison of changes in breast density categories and BMI over the ten years (2011–2020) at the starting age group.**

| | | Starting age (years), no. (%) | | | |
|---|---|---|---|---|---|
| | Total no. (%) | <40 | 40–49 | 50–59 | ≧60 |
| Changes in breast density category | | | | | |
| Decrease | 112 (25.3%) | 18 (33.3%) | 58 (24.2%) | 29 (24.2%) | 7 (25%) |
| No change | 286 (64.7%) | 30 (55.6%) | 161 (67.1%) | 76 (63.3%) | 19 (67.9%) |
| Increase | 44 (10%) | 6 (11.1%) | 21 (8.8%) | 15 (12.5%) | 2 (7.1%) |
| Changes in BMI | | | | | |
| Decrease | 69 (31.2%) | 3 (11.1%) | 38 (31.7%) | 23 (38.3%) | 5 (35.7%) |
| No change | 9 (4.1%) | 0 (0%) | 3 (2.5%) | 6 (10%) | 0 (0%) |
| Increase | 143 (64.7%) | 24 (88.9%) | 79 (65.8%) | 31 (51.7%) | 9 (64.3%) |

**Notes.**
BMI, body mass index.

In addition, lower BMI at the start tended to be associated with the higher breast density category.

Based on the data obtained, an ordinal logistic analysis was performed using the change in breast density category (three classes: decrease, no change, and increase) as the objective variable (Table 3). There was a statistically significant difference ($p < 0.005$) for the breast density category at the start and BMI. The coefficients for these two items were negative, indicating that the breast density category was lower over time with higher breast density at the start or higher BMI at the start. Odds ratios were 0.044 (95% CI [0.025–0.076]) for breast density categories at the start and 0.760 (95% CI [0.696–0.829]) for BMI. The odds ratio of BMI at the start was higher than the odds ratio of breast density at the start. Since

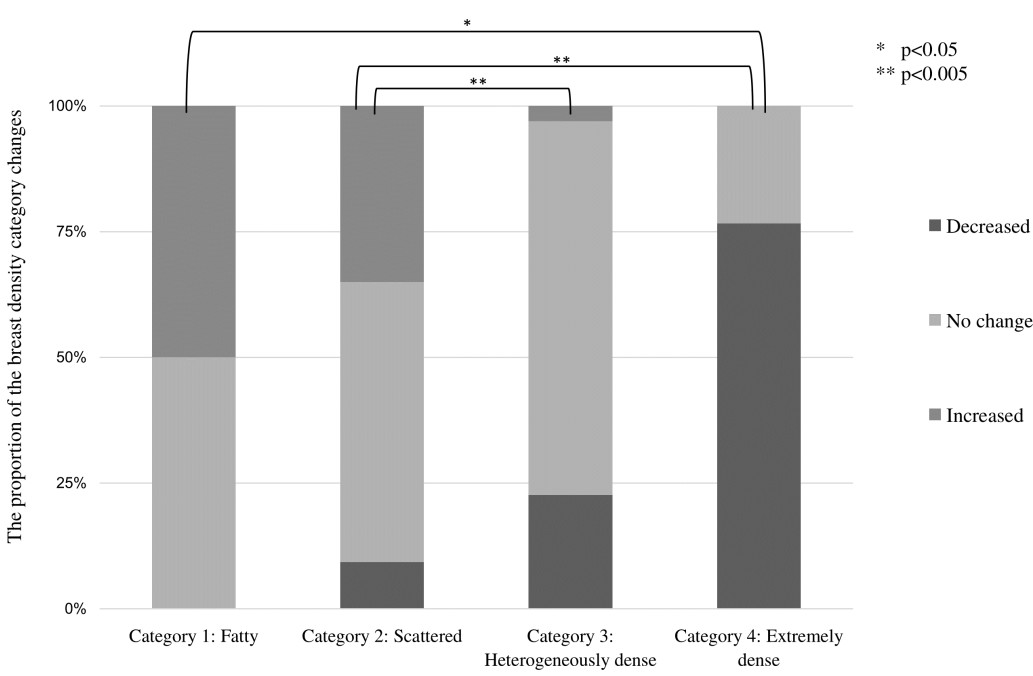

**Figure 4** Proportion of changes in breast density category (decrease, no change, and increase) compared by breast density category at the start of the study. Multiple comparisons by Fisher's tests for each breast density category showed significant differences. (* $p < 0.05$, ** $p < 0.005$).

BMI at the start was a continuous variable in the ordinal logistic analysis, a higher value of BMI at the start was more likely to decrease breast density, on holding other explanatory variables. There was also a statistically significant difference ($p < 0.05$) in the group with increased BMI and the group 40s at the start, and the coefficient was a negative value, suggesting that the increased BMI group and starting age in the 40s are also factors that cause the breast density category to lower. No statistically significant difference was found for the presence or absence of a history of benign breast or gynecological diseases.

## DISCUSSION

No change in breast density over the 10 years from the start of the survey accounted for the most significant percentage (64.7%), followed by a decrease (25.3%). The highest percentage of no change in breast density occurred in starting category 3, while the highest percentage of the decline occurred in starting category 4. Ordinal logistic analysis showed that breast density tended to decrease when the breast density category at the start was higher or the BMI value at the start was more elevated. The presence or absence of medical history did not affect the change in breast density.

In this study, breast density in category 3, the most common category, remained unchanged. The ordinal logistic analysis revealed that breast density tended to decrease in the 40s at the starting age. Warren et al. reported that around 80% of women had unchanged breast density over 6 years regardless of starting age (*Warren et al., 2019*). There was no change in mammary density in the majority of women of the study by *Engmann*

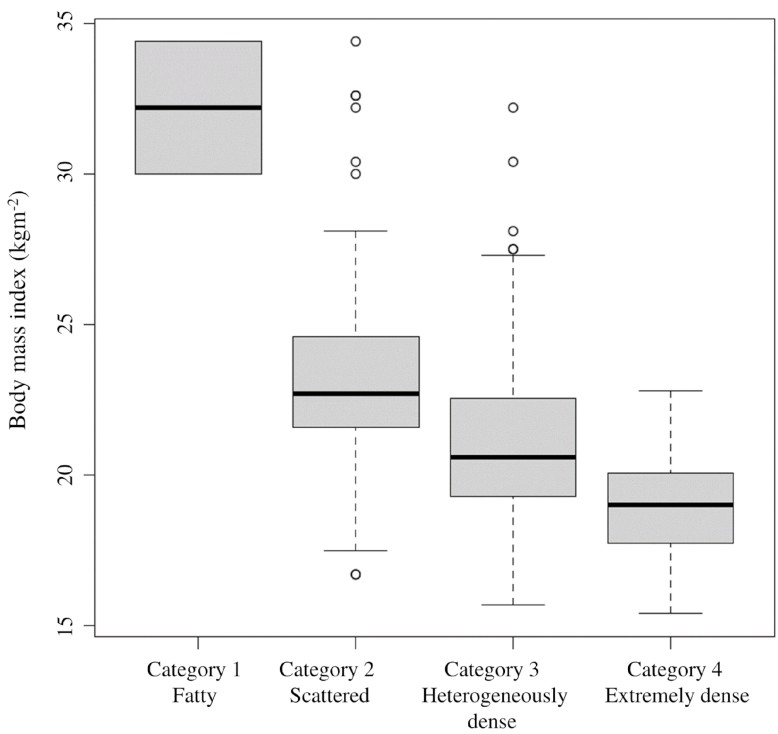

**Figure 5 Relationship between breast density category at the start and BMI at the start.** The 95% confidence intervals for the BMI values at the start are as follows: category 1: 26. 7–28.9; category 2: 23.4–23; category 3: 21.1–21.3; category 4: 18.3–18.8.

*et al. (2019)*. The study of *Kerlikowske et al. (2007)* also reported no change was most common in category 2.

As shown in Fig. 4, this study's most significant reduction in breast density occurred in women with a starting category 4. In Table 1, Category 4 accounted for 10% of all women, and most were under their 40s at the start. In the study by *Kim et al. (2020)* the mean age of women with non-dense breasts was 47.2 years, and the mean age of women with dense breasts was 39.2 years. The trend toward higher breast density in younger women and lower breast density with increasing age was age-related changes in breast volume density rather than breast density. Breast volume density is calculated by Volpara (Matakina Technology, Wellington, New Zealand), Quantra (Hologic, Bedford, Mass.), and other mammography analysis software. According to Warren et al., breast volume density decreased by approximately 11% over 6 years, and most women who experienced a decrease were dense breasts. The study of Engmann et al. compared pre-and postmenopausal women, and found a more significant reduction in breast volume density in women with premenopausal breast volume density of 54.3 cm$^3$ or greater (*Engmann et al., 2019*). Moshina et al. also reported a decrease in breast volume density of up to 10% over 6 years of age (Moshina et al.). Breast density in the majority remained unchanged, although some women's breast density decreased with age. Women with declining breast

**Table 3  Effect of baseline characteristics, changes in BMI, and the presence or absence of medical history on changes in breast density.** An ordinal logistic analysis was performed with the three groups of temporal changes in breast density categories as the objective variable. Explanatory variables are the five items; breast density at the start, BMI, age, changes in BMI, and the presence or absence of a history of benign breast or gynecological diseases. On mammograms, all women had no abnormality (including apparently benign disease).

| | | Estimate | Std. error | z value | Pr(>\|z\|) | OR | (OR 95% CL) |
|---|---|---|---|---|---|---|---|
| **Intercepts** | | | | | | | |
| Change in breast density | Decrease \| No change | −17.399 | 1.718 | −10.127 | 0.000(>0)[**] | | |
| | No change \| Increase | −12.825 | 1.571 | −8.163 | 0.000(>0)[**] | | |
| **Coefficients** | | | | | | | |
| Breast density at the start | | −3.127 | 0.282 | −11.098 | 0.000(>0) | 0.044 | (0.025–0.076)[**] |
| BMI at the start | | −0.275 | 0.045 | −6.13 | 0.000(>0) | 0.76 | (0.696–0.829)[**] |
| Change in BMI | No change | −0.295 | 0.559 | −0.528 | 0.598 | 0.744 | (0.249–2.228) |
| | Increase | −0.558 | 0.243 | −2.3 | 0.021 | 0.572 | (0.356–0.921)[*] |
| Age at the start | 40s | −0.723 | 0.365 | −1.979 | 0.048 | 0.485 | (0.237–0.993)[*] |
| | 50s | −0.707 | 0.403 | −1.756 | 0.079 | 0.493 | (0.224–1.086) |
| | >60 | −1.053 | 0.549 | −1.917 | 0.055 | 0.349 | (0.119–1.024) |
| Medical history | presence | 0.085 | 0.234 | 0.365 | 0.715 | 1.089 | (0.689–1.721) |

Notes.

[*] $p < 0.05$.

[**] $p < 0.005$.

BMI, body mass index.

density had higher breast density at the start. It was consistent with our findings in this study.

Over 10 years of the current study, the breast density category tended to decline from category 4, especially for women under the 40s. *Burton et al. (2017)* showed that breast area density, the ratio of the mammary gland area to the total breast area calculated from a mammogram, was lower in older women regardless of country or race. Therefore, the 40s women with dense breasts may lose their breast density over time and no longer have dense breasts. Although this study was conducted over 10 years, we believe that a 20-year study of women under 40s until they reach their 50s or older could provide a clearer picture of the effects of aging on changes in breast density. A longitudinal study in the UK demonstrated that BMI-adjusted density decreased with increasing age at screening (*McCormack et al., 2008*).

The impact of changes in breast density over time on breast cancer risk has been reported with different results. In the study by *Kerlikowske et al. (2007)*, which was a prospective cohort study of approximately 300,000 American women who had at least two mammograms between 1993 and 2003, women whose category decreased within 3 years compared to women whose class did not change have a reduced risk of developing breast cancer. The study by *Lokate et al. (2013)* points out that the more significant the decrease in breast density may affect breast cancer risk. The study by *Azam et al. (2020)* used the KARMA cohort data for Swedish women and compared changes in mammary tissue area within the breast; no significant difference in the hazard ratio for breast cancer risk was found for women with unchanged or increased mammary tissue area compared than women with decreased mammary tissue area (*Azam et al., 2020*). Although different

results have been reported on whether changes in mammary gland density over time affect breast cancer risk, it will be a point of interest in breast cancer screening.

During our study, 64.7% of women showed an increase in BMI. There was no age-related difference in BMI at the start, but the more significant the BMI at the start, the lower the breast density category. Results of the ordinal logistic analysis showed a strong trend toward lower breast density with higher BMI at the start ($P < 0.005$) and also a trend toward lower breast density categories with BMI increased over time ($P < 0.05$). Since percent breast density is determined by the ratio of adipose tissue to mammary tissue in the breast, it is assumed that women with higher BMI also have more adipose tissue in the breast, which would decrease breast density. This is consistent with the report by *Atakpa et al. (2021)* that breast volume density and BMI are inversely correlated. However, it has been pointed out that changes in BMI may change the apparent breast density. In contrast, the study by *Lam et al. (2000)* states that BMI and breast density independently affect breast cancer risk. In the study of Azam et al., BMI was one of the determinants of changes in breast density, women with BMI below 20 had lower breast density than those with BMI above 30 (*Azam et al., 2019*). Of the women in our study whose BMI had not changed since the start of the study, 56% also had no change in breast density. In addition, multiple comparisons by the Fisher test for BMI change showed no statistically significant differences among the three groups. Therefore, it is thought that the BMI value at the start can affect breast density more strongly than the change in BMI.

In the present study, there was no effect of a history of benign breast or gynecological disease on changes in breast density. Women with benign breast or gynecological disease accounted for 33.9% of the total. By age, the starting age of women with the medical history was 40 years or older; half in their 60s and older, but the number of subjects was small (14 cases). The facilities that cooperated in this study mainly provide medical examinations for workers. Therefore, we assume that the number of women in their 60s and older, which is the post-retirement age group, has decreased. In the study by Lokate et al., breast density decreased when hormone replacement therapy was given to women younger than 45 years (*Lokate et al., 2013*). Information on hormone replacement therapy was not collected in this study.

It is known that women with dense breasts who are at high risk for breast cancer have low detection rates on mammography, and other studies have been conducted in various countries using MRI and breast US as adjunctive tests (*Mariscotti et al., 2014*; *Ohuchi et al., 2016*). Multiple tests are expected to be time-consuming and financially burdensome for women. It should be noted that these tests were performed in high-density breasts because the stage of the disease is more likely to be advanced at the time of breast cancer detection. In treating breast cancer, *Sun et al. (2018)* reported that the average cost of treatment for breast cancer increases as FIGO staging increases. The present study suggests that women with dense breasts may also experience decreased breast density. Therefore, rather than recommending permanent multiple examinations for women with dense breasts, we believe that suggesting examinations based on breast changes will reduce the time and financial burden.

The present study was conducted on women who had continued examinations at a single facility. Since the facility conducted many workplace health examinations, the sample of women in their 60s and older was small, presumed to bias the target population. Future longer-term observations will need to consider differences in the birth year cohort. In addition, we used a self-developed program rather than commercially available software of mammogram analysis to adapt a retrospective study although a longitudinal study was conducted that followed the changes in individual women. For breast density measurement, most of the mammography analysis tools currently available use the raw data from the imaging to calculate the density, and this method was used in the study by *Azam et al. (2020)*. However, it was difficult to analyze past images stored in compressed format. In this retrospective study, we used an AI trained on images from the DDSM database to determine the breast's composition objectively. In practice of breast cancer screening, mammography image analysis software such as Volpara (*Engmann et al., 2019*; *Moshina et al., 2018*) and AI are used to determine mammary gland density or breast density (*Azam et al., 2020*). *Kling et al. (2022)* found that image classification models using Lobe yielded results comparable to those obtained with conventional machine-learning models. The mammograms used in this study were taken by two different devices. But mammographic images were taken at the same facility and had adjusted so that there were no significant differences in image quality between the two devices. It has been reported that the effect of differences in mammography equipment on breast density determination was small (*Damases, Brennan & McEntee, 2015*). Therefore, there would be no significant impact on the breast density measurement. One limitation of this study is that the images registered in the DDSM are digital images created by reading mammograms taken on analog film with a dedicated device. Compared to images currently used in clinical practice, the degree of blackening is low and contrast is poor. Therefore, there is a possibility that the composition of the breast will be judged low in the judgment of the model created in this study.

## CONCLUSIONS

This study was a retrospective, longitudinal study of breast density in women who had no abnormalities on mammography. Over the 10-year study period, breast density did not change in about 65% of the women, while breast density decreased by about 25%. Women with reduced breast density tended to have a higher breast density or a higher BMI at the start of screening. This effect was more pronounced in women with the 40s at the start. Women with these conditions may experience changes in breast density over time.

## ACKNOWLEDGEMENTS

I would like to thank the Medical Corporation Shin-ai, and especially the Radiology Department for their cooperation in collecting mammogram.

### Funding
The authors received no funding for this work.

### Competing Interests
Kazuhiro Maeda, Hiroyuki Ono, Kyoko Iwasaki and Kazuhiro Mori are employed by Medical Corporation Shin-ai. Seiichiro Kamimura are employed by Chiyukai Shinkomonji Hospital.

### Author Contributions
- Aiko Ohmaru conceived and designed the experiments, performed the experiments, analyzed the data, prepared figures and/or tables, authored or reviewed drafts of the article, and approved the final draft.
- Kazuhiro Maeda performed the experiments, authored or reviewed drafts of the article, provide mammography image data and patient information, and approved the final draft.
- Hiroyuki Ono performed the experiments, authored or reviewed drafts of the article, provide mammography image data and patient information, and approved the final draft.
- Seiichiro Kamimura performed the experiments, authored or reviewed drafts of the article, provide mammography image data and patient information, and approved the final draft.
- Kyoko Iwasaki performed the experiments, authored or reviewed drafts of the article, provide mammography image data and patient information, and approved the final draft.
- Kazuhiro Mori performed the experiments, authored or reviewed drafts of the article, application to the Ethics Committee,Provide mammography image data and patient information, and approved the final draft.
- Michiaki Kai conceived and designed the experiments, analyzed the data, prepared figures and/or tables, authored or reviewed drafts of the article, and approved the final draft.

### Human Ethics
The following information was supplied relating to ethical approvals (i.e., approving body and any reference numbers):
  This study has been reviewed by the Ethics Committee of the Medical Corporation Shin-ai.

### Data Availability
  The raw data are available in the Supplementary File.

## Supplemental Information

Supplemental information for this article can be found online at http://dx.doi.org/10.7717/peerj.14836#supplemental-information.

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
