# Peer review of "Age-related change in mammographic breast density of women without history of breast cancer over a 10-year retrospective study"

_PeerJ, doi:10.7717/peerj.14836_

## Round 0.1 · original submission · Major Revisions

We now have reached a total of 3 independent peer reviews and myself. I have reached a decision. Before being accepted for publication your manuscript requires significant revisions. Please, consider all the reviewers' comments and re-submit your manuscript and responses to reviewers as soon as possible. Please, let me know in case you require more time.

Please, also ensure that your statistical analysis and inclusion/exclusion criteria are clear, comprehensive, and properly applied.

Please, note that copyediting is not provided as a standard publication service. Please ensure the language in this submission is clear and unambiguous, grammatically correct, and conforms to professional standards of courtesy and expression.

Reviewer 1 has suggested that you cite specific references. You are welcome to add it/them if you believe they are relevant. However, you are not required to include these citations, and if you do not include them, this will not influence my decision.

Reviewer 1 ·

Basic reporting

The paper is clear and the research question is interesting.
The literature and references need additional work. Please see the below comments.

Introduction:

* Line 59: “breast density has recently attracted attention” - This is not entirely correct because the first papers published back in 1990s and it is very well known and established risk factor for breast cancer for many years. Please remove this claim.
Warner E, Lockwood G, Math M, Tritchler D, Boyd NF. The risk of breast cancer associated with mammographic parenchymal patterns: a meta-analysis of the published literature to examine the effect of method of classification. Cancer Detec Prev 1992;16:67-72.

Boyd NF, Lockwood GA, Byng JW, Tritchler DL, Yaffe MJ. Mammographic densities and breast cancer risk.
Cancer Epidemiol Biomark Prev 1998;7:1133-44.

* Line 61: Please define mammographic density properly and refer to the highly cited papers on mammographic density and risk of breast cancer.

* Line 69: please remove (OR=1).

* Line 74: This is not entirely correct “reported that the average mammary gland density”. Warren et al reported average volume breast density (VBD) decreased by 11% over six years not “average mammary gland density”. Please remove it and use the correct term.

* Please note to also refer to the papers which measured mammographic density change per year

Shadi Azam, Arvid Sjölander, Mikael Eriksson, Marike Gabrielson, Kamila Czene, Per Hall, Determinants of Mammographic Density Change, JNCI Cancer Spectrum, Volume 3, Issue 1, March 2019, pkz004, https://doi.org/10.1093/jncics/pkz004
Boyd N , Martin L, Stone J, Little L, Minkin S, Yaffe M. A longitudinal study of the effects of menopause on mammographic features. Cancer Epidemiol Biomarkers Prev. 2002;11(10 Pt 1):1048–1053.

Experimental design

Materials & Methods
* Please mention if this is a retrospective cohort study.
* Please define what are the medical history of breast and gynecology
* Change in BMI, how did you measure that? Absolute or relative change?
Line 96: Please remove this “undergone 51,573 mammography examinations”.
Line 101: why women with the age of 25 had mammography in this cohort?
* Any information on family history of breast cancer?
Line 112: “Determination” please remove and replace by “Breast density assessment”
Line 136: Did the authors used average of left and right density?
Line 138: How the change over time was measured? Did you measure relative or absolute density change?
Line 153: Why the author did not use the average breast density over the left and right breasts?
Line 185: Who the 3 categories of mammographic density change defined? “Breast density category (three classes: decrease, no change, and increase)”.
Line 196: I suggest excluding the women with previous breast cancer, this is because the breast cancer treatment such as surgery, chemotherapy, and radiation therapy significantly affects mammographic density.
Line 203: “. Ordinal logistic analysis showed that breast density 204 tended to decrease when the breast density category at the start was higher or the BMI value at the start was more elevated.” This is very much expected because the authors measured absolute mammographic density change over time. However, it is mostly suggested to measure relative mammographic density change. You can read more on this here:
Shadi Azam, Arvid Sjölander, Mikael Eriksson, Marike Gabrielson, Kamila Czene, Per Hall, Determinants of Mammographic Density Change, JNCI Cancer Spectrum, Volume 3, Issue 1, March 2019, pkz004, https://doi.org/10.1093/jncics/pkz004

Validity of the findings

The results are valid and interesting findings, I have some additional comments for the discussion.
Line 252: which study is this? “By Asam et al.” the year is missing, is it “Azam et al.”?
Line 261: This is not entirely correct , “Since breast density is determined by the ratio of adipose tissue to mammary tissue in the breast, it is assumed that women with higher BMI also have more adipose tissue in the breast” – the statement is only correct for percent mammographic density, however, dense area is not affected by BMI. Therefore, please replace breast density with percent breast density.
279: “breast density decreased when hormone replacement therapy was given to women younger than 45 years” Please provide the exact paragraph and sences regarding this finding in this study (Lokate et al., 2013).
305: Please replace “determination of breast density” by “breast density measurement”.

Reviewer 2 ·

Basic reporting

The authors present a clear and professional English language throughout the paper. There are some specific issues with some scientific terms that I addressed in the following comments and that the authors should consider.
There is a correct reference of relevant literature, however, it would have been good to see some specific topics more developed (for example the relationship between breast cancer risk and breast density. I specify what I mean with more examples further on - see general/specific comments).
In terms of structure, the paper is in line with PeerJ standard, having an Abstract within the 500 word limit, well-organized, describing in a concise manner each section that composes the paper. However, “Methods” should be substituted by “Materials and Methods”. Besides that, when the authors state “the ordinal logistic analysis was performed”, it should be clear why it was performed - To verify what?
The remaining of the paper is organized as expected: Introduction, Materials and Methods, Results, Discussion, and Conclusion.

Figures and Tables are relevant and a very good complement to what is said on the text. However I have some concerns: there can not be “new” relevant information on the figures. If it is relevant it needs to be explicit in the text. Besides that, the font used for Figures and Tables should be uniform. The labeling of the figures is too generic. A brief explanation of what is being observed could be a good complement that would improve the reader's experience. Same rationale for tables.
In relation to raw data, there are some concerns related to the provided csv file. When analyzing figure 5, it is possible to verify that there are several inclusion criteria: having mammography recorded more than eight times (in the text it is said “at least” - line 101); BMI being recorded more than eight times; medical history recorded; mammogram from 2011 saved digitally. This resulted in 221 included cases.
However, when looking at the csv files, some questions arise:

- For example, for ID=1, BMI was only recorded eight times - not “more than eight” - was this case included? If yes, why?
- For ID = 5, even though it underwent mammography more than eight times, BMI was only recorded eight times. Was this case included? If yes, why?
- If I understand correctly, ID’s = 2,4, 5, 7, 8, 10, … (there are more), don’t have a medical history (value=0 in the “history” column). Were these cases included? If yes, why?

I gave here some examples of the main concerns related to the provided data. As for the first two points, this can be countered by changing “more than eight times” to “eight times or more”, in the inclusion criteria. However, screening the entire dataset is recommended to verify if the new inclusion criteria is met.
The third point is of bigger concern, because there are several cases with value = “0” in the history column, and this variable is studied in the ordinal logistic regression table. It is important to verify if all the 221 included cases had medical history or not. If not, it is important to state how many cases were used for the ordinal logistic regression. Once again, the authors are recommended to screen the entire dataset, correct the mentions on the text, and adapt - as seen suitable - the inclusion criteria.


General Comments about the Introduction section:

The Introduction section starts by giving a wider view of the breast cancer paradigm in terms of incidence rate and mortality, both in Japan and in Western countries. It identifies breast density as a risk factor for this disease, giving literature references that support the said identification. The authors even point to case-control studies in Japan that identify the relationship between high breast density and breast cancer. These references are highly important since they show that breast density can be considered as a risk factor for the population examined in the proposed study.
However, it is not clear if one should expect that breast density decreases with age. Only two studies are presented. One claims that breast density did not change in 80% of the analyzed women, the other points to a decrease of 11% over 10 years (not being clear if it occurs in all analyzed women). The authors should explore the literature and deepen the analysis done from lines 72-77.

The reference to the fact that mammography has a low detection rate for dense breasts is important once it shows that standard mammography might not serve all women. However, the authors do not explore these mammography flaws nor do they investigate the possibility of adapted screening based on risk factors such as breast density - the sentences present in lines 81-84 flow in that direction but without a clear conclusion. The authors may consider deepening this discussion a little bit.
At the end of the Introduction section, the authors present their study goal: “Clarify the proportion of four categories of breast density (...) among Japanese women”, which is perfect. However, the authors did more than “investigate the changes in breast density over time”. The authors should present a more concludent aim. For example: “investigate the changes in breast density over time, and understand which factors may cause those changes.”


Specific Comments about the Introduction section:

Line 58. Not “recently”. There are studies from the 70/80/90’s that link breast density to breast cancer risk. (https://pubmed.ncbi.nlm.nih.gov/918102/; https://pubmed.ncbi.nlm.nih.gov/179369/ ; https://pubmed.ncbi.nlm.nih.gov/2025849/ )

Lines 61-62. For better understanding, either the definition of the category or the category itself should appear between parentheses, i.e., “Category 3 (heterogeneously dense) …” or “heterogeneously dense (Category 3)”. Analogous for category 4.

Line 69. “They also…”. It is not clear to whom it refers. If it is referent to the immediately previous analyzed research, then it would be more appropriate to say something like “The authors” instead of “They”. On the other hand, if it refers to the papers that are cited at the end of the paragraph, the use of “They” makes no sense and therefore the beginning of the sentence should be rewritten.

Line 72. Stating that there are differences in proportions of each of the four categories is not enough. How are these differences? And how are the age groups defined?


Lines 73-74. What is the message that it is trying to be passed? The information seems slightly contradictory. One study states that in 80% (a big percentage) of the women the density remains unaltered. The other states that there is a decrease of 11% over 10 years - does this happen to all women in the study? It is expected to see a decrease in every woman, or just in 20% of them (considering the Warren study)? The authors should consider clarifying this paragraph, once it is of extreme importance for the aim of the proposed study.

Line 81. “Test” should be replaced with “imaging techniques” or an analogous expression.

Line 89. “Test” should be replaced with “type of screening” or an analogous expression.

Experimental design

This is an original primary research in the field of Medical/Health Sciences, so it is within the scope of the journal.

The research question is defined at the end of the introduction. It is not exactly a question, as it is more of a goal - to clarify the proportions of four categories of breast density and to investigate changes in breast density over time. This objective is relevant - it is known that breast density is a risk factor for the development of breast cancer so, understanding how this variable affects different age groups and how it changes across time could be important to adapt/change screening routines. Nonetheless, it would be important to see at least a sentence after the objective presentation that describes which gap in the literature this research fills.

The investigation follows a rigorous process. The use of an Artificial Intelligence model for breast density classification is state-of-the-art, which is very positive. However, as I describe further on, it needs some clarification related to the testing phase. Otherwise, the methodology will be difficult to replicate. The analysis of the data is conducted through a statistical analysis of the data that should be a little bit more developed, for better understanding. The paper could benefit from a brief explanation of the ordinal logistic regression model and the possible interpretations.

Related to the ethical concerns, the authors provided a privacy police that was disseminated to both staff and related parties. The privacy policy states what data is being collected, and how it can be used (and by whom). As far as I understand, all the participants agreed to it (Line 109). This study was presented to an ethics committee that approved it.


General Comments about the Materials and Methods section:

The authors start with a description of the collected data. Although there is clarity about the number of analyzed subjects, please consider the specific corrections presented below (specific comments in order to improve readability.)

If possible, I would recommend to state where the examinations took place. There is a reference to both data protection (anonymity) and informed consent which is very positive.

The authors clearly describe how the decision-making model for breast density categorization was developed, which is very important for replication purposes. However, it is not clear what the authors mean about the accuracy of the model. They state that the model was tested with a “testing function” but do not clarify with what data.

Figure 2 includes a box with the following text: “Improve model by test results feedback”. However, I do not see any mention in the text about this step. Authors should consider explaining how this feedback works.
The color code in Figure 3 is not coherent. For visual interpretation, the tonality increases from category 2 to category 3, and then from category 3 to category 4, which is perfect. However, category 1 has a tonality very close to category 4, which is not ok, it should be lower than category 2. Please consider revising the color scheme.

The comparisons about breast density distributions across different groups is very interesting, nonetheless, a comparison to what was expected would improve the quality of the work. For example, it was expected that the group with most category 4 breasts was the under 40’s? What does the literature say? How is the expected breast development as a woman ages?

Specific Comments about the Materials and Methods section:

Lines 98-103 – Changing the way that the dataset construction is presented could improve readability. I recommend that you state the criteria at the beginning of the paragraph and then, as you did, say how many women did (or did not) match the said criteria. Besides that, please try to clarify what you mean about “medical history”. It is only clarified in Figure 1 (BMI, breast, and gynecology history), and in the next paragraph, but it could be important to understand immediately what it is meant by that.

Line 113. Instead of “udder” use breast/breast tissue.
Line 120-121. Please improve the use of English in this sentence.
Line 127-129. Please conclude the sentence.
Line 128. ACR acronym was not defined.
Line 131. “Accuracy improved to 97%”. Improved in relation to what?
Line 144-147. Please consider deepen the explanation about the ordinal logistic regression. What it is, what it does, and what interpretations can be driven by it.

Validity of the findings

After the author's answer to the posed comments, meaningful replication will be encouraged through a well-defined rational. As commented previously, the authors should clearly identify which gaps in the literature are they filling with each of their presented aims. For comments about data robustness please refer to the comments made in “basic reporting”. The conclusion section should be longer than four sentences: besides developing the describing of their findings it is important that those sentences are connected to the previously defined aims and goals, proving that they were met. Besides that, a sentence that explains how the obtained results (or the study itself) helped to fill a gap in the literature should also be present.

General Comments about the Results and Discussion sections:

The authors start the Results section by describing the dataset information in terms of age and BMI. This information is not a result and it should be part of the Materials and Methods section.

Table 2 shows the variations in breast density between the final and the first year. It could be interesting to understand how abrupt these variations were, i.e., slight variations across the years that resulted in great variations when comparing the extreme years or an abrupt variation from one year to another? Besides that, all decreases/increases are grouped together, but an increase from category 2 to category 3 certainly has a different meaning than an increase from category 1 to category 4? Same rationale for decreases.
When evaluating the correlation between initial BMI and breast density category, statistically significance analysis should be deepened in order to clarify the relevance of the findings. Also, I have some concerns about the relationship between breast density and BMI, please see “specific corrections” for more details.

The analysis done on table three (lines 184-197) is not perfect. The objective variable is well defined however, I have some concerns about the following sentence on line 187: “There was a clear, statistically significant difference (p<0.005) for each explanatory variable for breast density category at initiation and BMI”. What do you mean by “each explanatory variable”? As I see when analyzing table three there is significance for “breast density category at initiation” and for “BMI”, but not for “each explanatory variable”. Please clarify what you mean by explanatory variable and point out that the significance is observed for “BMI at the start” and not simply “BMI”.
I find it very difficult to understand that an elevated BMI at the start results in a tendency to decrease breast density, because, when analyzing Figure 5, the elevated BMI values (>25kgm-2) are related to the lowest density class. Please deepen the analysis done to these results and clearly describe the rationale behind each driven conclusion.

Considering the Discussion section, the literature analysis and its comparison with the obtained results done between lines 216-227 is very interesting and it shows that the performed research is in line with what can be found in the reviewed literature. However, some studies present a considerable reduction in breast density. It would be interesting for the authors to compare those findings with the ones present in their study.

When describing results that analyze the relationships between density and cancer risk (lines 237-249) , the authors should not simply say “breast density changes” - it is important to understand how these changes are, i.e, does the density increase or decrease?
I fail to see the relevance of the paragraph that starts at line 250.

As for the Conclusion sections, comments were already made.

Specific Comments about the Results and Discussion sections:

Line 172: It should be “the largest proportion of breast density change .. “ ?

Line 182: “In addition, lower starting BMI tended to be associated with the lower breast density
Category.” Is this correct? When analyzing Figure 5, lower BMI (15-20 kgm-2) seems to be associated with higher categories - specially 3 and 4.

Line 189-191: If I understood correctly, it is said that the density category tends to decrease with higher breast density at start and higher BMI at the start. In the same manner in both of the cases? Or is it more clear in one of the variables? Please consider analyzing that if you find it relevant.

Line 194: “suggesting that the increased BMI group and starting age in the 40s are also factors that cause the breast density category to lower” - So, if an increased BMI is a signal that breast density is set to lower, how come that - as said in line 182 - lower BMI is associated with lower breast density?

Line 214: “As shown in Table 3, the most significant reduction in breast density in this study occurred in women with a starting category 4”. Where exactly in table 4? I see that there is a significant value in a variable called “Breast density at the start”. It is not clear, however, that that variable is in any form related to category 4 - It needs to be evident in any data that it is shown.

Line 258: “but the more significant the BMI at the start, the lower the breast configuration” - It should be clear, but it is understandable that a significant BMI means a higher BMI. However, what does a “lower breast configuration” mean? Please state clearly if this means less glandular or fatty tissue. Or if you are trying for some different meaning.

Line 262: “is assumed that women with higher BMI also have more adipose tissue in the breast, which would decrease breast density”. Exactly! So why did you say in Line 182 that a lower BMI is associated with lower density categories?

Additional comments

Dear authors,
Thank you so much for the submission of your very valuable work.
Further on you will find some general comments, suggestions and concerns with your paper. Some specific corrections are also addressed.
Once again I thank you very much for the effort, time, and professionalism that you put in your paper.

Reviewer 3 ·

Basic reporting

The issue of breast density (BD) has classically been considered to significantly affect the radiologist's ability to assess mammography. More recently, its possible role as a risk factor for the incidence of breast cancer has been considered with greater attention, especially in a screening environment – with a low a priori prevalence rate – and as a factor that favors “technical” interval cancers”.

Experimental design

The present study aims to quantify the evolution of BD in women without breast cancer (in the different moments – 8? - of obtaining the mammogram) with the beginning of regular exams from 25 to 65 years old. Note that line 109 refers as “patients” to women who are not (at least for breast cancer). On the other hand, the reference to “history of previous breast cancer” (lines 195, 196, 197, 275, 276) seems inappropriate, as it should have been (was it?) considered as an exclusion criterion.

My biggest criticisms of the article are:
1. There is no clear division (in the analysis) between the different age cohorts at study entry. They are very diverse cohorts, with different reproductive and social lifestyles, and lacking, for example, a factor that could be fundamental – menopause. BMI single use seems a bit limited
2. Eventually, an age-cohort like analysis (inserted into an age-period-cohort like analysis) could be interesting to differentiate period and age evolution. Would be better suited to a more robust descriptive analysis
3. The non-use of an already standardized commercial BD quantification product could have been used to improve comparability. There exist software that evaluate films “for processing” and “for presentation”

Validity of the findings

The overall number of cases is small (as in the group <40 and >60), making us immediately suspect that there may be bias and chance.
So the external validity seems to be small

Additional comments

Therefore, I think that although it is an interesting study, it should have some corrections and clarifications to be considered for publication.

---

## Round 0.2 · Major Revisions

Methodological concerns and of interpretation remain. Please, address this and the reviewers' comments.

Reviewer 1 ·

Basic reporting

The manuscript improved after the major revision. I have no further comments.

Experimental design

The research questions are relevant and the study is well-designed.

Validity of the findings

The study results are valid and conclusions are well stated now.

Reviewer 2 ·

Basic reporting

Introduction section:

1.
Recall the comment: “However, it is not clear if one should expect that breast density decreases with age. Only two studies are presented. One claims that breast density did not change in 80% of the analyzed women, the other points to a decrease of 11% over 10 years (not being clear if it occurs in all analyzed women). The authors should explore the literature and deepen the analysis done from lines 72-77.”

The authors responded with: “Lines 66-72 of the revised paper were added.”

These lines are not related to the mentioned problem. I suggest that the authors revise lines 79-84, and explain based on the found literature if it should be expected that breast density decreases with age.


2.
Recall the comment: “Line 72. Stating that there are differences in proportions of each of the four categories is not enough. How are these differences? And how are the age groups defined?”
The authors responded with: Line 72 was incorrect to begin with, and corrected it.

The sentence (now line 79) was changed but the question remains, how are these four categories defined and how do they relate with age groups?

Experimental design

After the asked clarification about the testing of the algorithm, to which the authors responded on lines 147-150, some concerns arise regarding the methodology used: “images for which Lobe’s model made an incorrect decision were trained to make the correct decision again” (lines 148-49). Does this mean that images used for training were also used for testing? If yes, does the 97% accuracy result have some relevance? The authors should deeply review what was done here and try to clarify exactly what was the methodology used. If there are images both in training and testing, not only the accuracy results are irrelevant, as the developed model probably won't be able to generalize its performance to another dataset (or even for never seen images from the same dataset). Don’t leave any room for doubt. Clearly explain how your methodology was: image division into training and testing, re-training process, and accuracy evaluation.

Validity of the findings

The readability of lines 207-210 needs to be deeply improved in order to understand the conclusions being driven. It is better than it was in the first version but it is still very difficult to understand the message that is trying to be transmitted.
As far as I understand, it is said that an Higher BMI at the start led to a lowering of density category over time: how is this possible if figure 1 shows that an higher BMI at the beginning is associated with the lowest possible category.

---

## Round 0.3 · Minor Revisions

Dear authors,

Thank you for your revised version. Before publication, please address in particular your writing and rationale in order to improve the readability and clarity of your article.

Please, revise the reviewers' earlier comments for guidance. Thank you.

---

## Round 0.4 · Minor Revisions

Dear authors, thank you for your reviewed manuscript. While you addressed most concerns and questions, a few remain, namely about any bias and consequent incorrect conclusions. Please review the reviewer's comments. I suggest revising and stating clearly your limitations and how the model was optimized! Did you use poor-quality images? Or did "it" relearn the images? What were the criteria to include or exclude images? Why? Clearly state your rationale for both readers to understand and researchers to validate your method, in case need be. Based on this, is your classifier able to properly analyze new data or not? The language style is also important, please consider it while revising.

Reviewer 2 ·

Basic reporting

No comment

Experimental design

Dear authors,
I remain with doubts concerning the training of your model.

Line 149: "were lousy in quality and location or confusing to the model" is what it is said about the removal/modification/retraining of images from the dataset. It needs to be clear what does it mean to "confuse the model". While I can understand that images with low-quality can disrupt the learning of the model, you need to clearly state what it is meant by "confusing".

I recommend that you clearly state the criteria that led to image removal/modification from the training dataset, and the rationale behind it. Otherwise, the random removal of "confusing" images could bias the classifier performance - meaning that it might not be able to generalize.

Validity of the findings

No comment

---

## Round 0.5 · accepted · Accept

Dear authors, congratulations on finally getting your manuscript above cited accepted for publication on PeerJ. I attach a pdf highlighting a couple of typos and the recent changes. Please, revise eveything, including figures and tables to confirm that all the information matches, and that there are not issues of any sort. The editorial office shall be in contact with you in case anything else is needed. Many thanks.